

# Detection of flooding by overflows of the drainage network: Application to the urban area of Dakar (Senegal)

Laurent Pascal Malang Diémé[1], Christophe Bouvier[2], Ansoumana Bodian[1] and Alpha Sidibé[3]

[1] Laboratoire Leïdi "Dynamique des Territoires et Développement", Université Gaston Berger (UGB), Saint Louis, Sénégal
[2] IRD, UMR 5151, HSM, Univ. Montpellier, CNRS, IRD, Montpellier, France
[3]DPGI "Direction de la Prévention et de la Gestion des Inondations au Sénégal", MEA, Sénégal

*Correspondence to*: Laurent P. M. Diémé (dieme.laurent-pascal-malang@ugb.edu.sn)

**Abstract.**

With the recurrence of flooding in African cities, there is growing interest in the development of sufficiently informative
tools to help characterize and predict overflow risks. One of the challenges is to develop methods that strike a compromise
between the accuracy of simulations, the availability of basic data, and the shortening of calculation times to be compatible
with real-time applications. The present study, carried out on the urban outskirts of Dakar, aims to propose a method capable
of modeling flows at fine resolution (5m$^2$), over the entire area, and providing a rapid diagnosis of how the drainage network
is operating for rainfall intensities of different return periods, while taking urban conditions into account. Three
methodological steps are combined to achieve this objective: i) determination of drainage directions, including modifications
induced by buildings, artificial drainage and storage basins, ii) application of a hydrological model for calculating flows at
the outlets of elementary catchment, iii) the implementation of a hydraulic model for propagating these flows through the
drainage network and a storage model for retention basins. The network overflow points are calculated as the difference
between the calculated flows and the network's capacity to evacuate them. Simulation results show that the stormwater
drainage network is capable of evacuating runoff volumes generated by rainfall with a low return period (10 years), but
seems to overflow for rainfall with a rare frequency (100 years), with overflow rates sometimes exceeding 18 m$^3$/s. The
model, built on the ATHYS modelling platform, also provides boundary conditions for applying more complex hydraulic
models to determine the local impact of drainage network overflows on limited areas.

## 1 Introduction

African cities are frequently subject to flooding (Yengoh et al., 2017; Tazen et al., 2018; Sy et al., 2020; Barau and Wada,
2021), which results in significant socio-economic, health and environmental damage (Miller et al., 2022a; Sakijege and
Dakyaga, 2023). The current trend towards more intense rainfall (Taylor et al., 2017; Bichet and Diedhiou, 2018; Nkrumah
et al., 2019; Klutse et al., 2021), attributed to climate change (Panthou et al, 2018; Chagnaud et al., 2022) and the very rapid
dynamics of urbanisation (Sène, 2018; Williams et al., 2019; Yuan et al., 2023), are expected to increase the recurrence of



urban flooding (Gaisie and Cobbinah, 2023). This is a major source of concern for political decision-makers and city
dwellers (Moulds et al., 2021) in these African conurbations, where the gap between adaptation needs and existing tools is
wide (Nkwunonwo et al., 2020; Miller, et al., 2022b).

In response to growing adaptation needs (Kreibich et al., 2017; Mashi et al., 2020), interest is being shown in flood
characterization (Coulibaly et al., 2020) and forecasting (Chen et al., 2015). In this respect, the scientific literature reports on
several methods implemented, in urban environments, to provide flood assessment and mapping (Henonin et al., 2013;
Agonafir et al., 2023). The simplest methods, without introducing simulations of runoff formation, rely on the topographical
characteristics of the territory to give a first local estimate of flood risk by accumulation of water at low points (Pons et al.,
2010; Dehotin et al., 2015; Zheng et al., 2018). As for the 1D hydrological and hydraulic modeling approach, well
established in the literature (Zhu et al., 2016; Rabori and Ghazavi, 2018; Sidek et al., 2021; Chahinian et al., 2023), it is
applied to simulate stormwater drainage network performance (Meng et al., 2019; Pla et al., 2019). Modeling platforms such
as SWMM (Rossman, 2015; Rabori and Ghazavi, 2018) or InfoWorks ICM are 1D simulation tools applied in urban
environments (Rubinato et al., 2013; Sidek et al., 2021). However, this type of modeling, which is essentially one-
dimensional, does not provide spatial propagation of overflow water (Mark et al., 2004). However, this type of modeling,
which is essentially one-dimensional, does not provide the spatial propagation of overflow water (Mark et al., 2004). This
aspect is taken into account by 2D models such as Mike Urban (DHI, 2021). The accuracy of the simulations they can
provide on the spatial propagation of surface flows is limited by their numerical complexity, to which is added the fine
quality of the data (fine topographic mesh, physical and urban characteristics) required for their parameterization (Costabile
et al., 2020; Zanchetta and Coulibaly, 2020). These 2D or coupled 1D-2D models (Martínez et al., 2018; Bulti and Abebe,
2020; Li et al., 2022) require substantial computing resources, long calculation times and are difficult to apply over large
areas or for real-time flood forecasting studies (Rosenzweig et al., 2021). Today, we are also witnessing the emergence of
increasingly used AI (artificial intelligence) / ML (machine learning) machine learning techniques (Mosavi et al., 2018;
Darabi et al., 2019), which have the potential to provide flood mapping through model training (Mosavi et al., 2018; Darabi
et al., 2019; Parvin et al., 2022; Taromideh et al., 2022). Their application can be challenging as they generally require a
large amount of data (meteorological, hydrological, topographical) to be integrated for training, in order to improve accuracy
and achieve good model performance (Bentivoglio et al., 2022).

In urban environments, one of the main factors influencing the choice of an appropriate modelling approach is data
availability and the flooding context (Henonin et al., 2013). For the African context, where the availability of detailed data is
rare, the challenge is to implement alternative solutions by finding a compromise between the availability of basic data, the
reduction of calculation times and the accuracy of flood simulations (Chahinian et al., 2023). The aim of this study is to
propose fine-resolution ($5m^2$) modelling of flows and overflows from drainage and storage network on the scale of the
Dakar's urban periphery with short calculation times (5mn) compatible with real-time applications. The proposed



methodological approach follows three main stages: (i) the reconstruction of urban drainage directions, taking into account the modifications caused by the various urban developments (buildings, artificial channels and retention basins), using algorithms developed for this purpose, (ii) calculation of flows in project mode, using a parsimonious hydrological model
(SCS-LR) adapted to the local context, which in particular integrates the density of urbanisation on the scale of small basins, combined with (iii) a 1D hydraulic model for propagating these flows through the drainage network and a storage model for retention basins. The overflow points in the drainage network are identified by the difference between the maximum flows produced and the network's capacity to evacuate them. This work is structured in four parts. First the study area is described, then the data and the detailed structure of the method are presented, followed by the model parameterisation strategy and
finally the results and a discussion highlighting possible improvements, before concluding.

## 2 Study area

Dakar (Fig. 1), the capital of Senegal concentrates almost ¼ of the country's total population on just 0.3% of the national territory (ANSD, 2013). Its urbanization took place rapidly, in the space of a few decades, largely fuelled by the rural exodus (Lericollais and Roquet, 1999) following the drought of the 1970s (Nicholson et al., 2000). This resulted in rapid population
growth and, with the establishment of the network of roads to facilitate urban mobility (Ndiaye, 2015), the land reserves of the distant eastern periphery have been invaded and densely urbanised (Lessault and Imbert, 2013). This is sometimes achieved through (i) the infilling of former drained wetland depressions (Sène et al., 2018), (ii) self-occupation practices, without taking into account the topography of the land or the installation of rainwater drainage structures (Ndiaye, 2015).

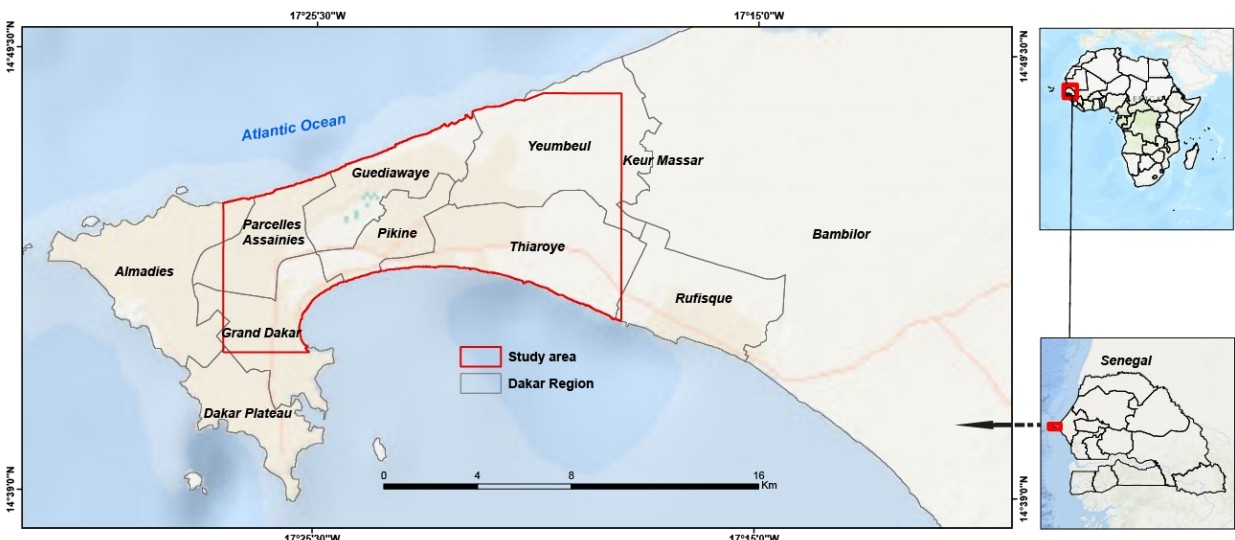

**Figure 1: Location of the study area**



From the 2000s onwards, there was a return of rainfall (Sene and Ozer, 2002; Bodian, 2014; Nouaceur, 2020) that caused a series of floods in Dakar (Bottazzi et al., 2018). Since then, these floods have occurred with near-annual frequency, and some are particularly disastrous with impacts prolonged over time (Hungerford et al., 2019). In 2012, for example, they had a serious impact on the population, resulting in 26 deaths (Sané et al., 2016) and causing pandemics such as cholera and malaria (Magny et al., 2012; Sambe-Ba et al., 2013). One of the government's responses was to set up a vast programme, including the Stormwater Management and Climate Change Adaptation project (PROGEP), which aimed to build drainage networks linked to storage basins to minimise the risks (Diop, 2019). One of the current challenges for urban management, in a context of increasing intense rainfall, dense urbanisation and infrastructure development, is to develop effective and robust tools to support flood assessment and decision-making.

## 3 Data and method presentation

The detailed methodological approach followed is structured in seven successive stages. The first is (i) the construction of the natural drainage topology modified by urban objects, (ii) on which is based the division of the urban area into small elementary catchment areas and the extraction of the associated hydrographic network. Then (iii) a hydrological production and routing model is applied to calculate the hydrographs at the outlets of the elementary basins. These hydrographs are (iv) propagated in the storm drainage network by a 1D hydraulic model and (v) their conservation in the retention basins is managed by a storage model. Finally, (vi) project rainfalls are constructed from local Intensity-Duration-Frequency (IDF) curves and injected in the model, which is then (vii) implemented to detect, over the entire study area, the overflow points of the drainage and storage network according to different levels of severity and urban density. The entire processing chain and the associated data are presented in detail in the following sections.

## 3.1 Construction of the drainage topologie

The construction of the drainage topology, aimed at reconstructing the modified drainage directions of run-off water, is a prerequisite to the implementation of the overflow point detection model. This topology construction methodology was previously applied to the study area (Diémé et al., 2022), using spatial data from the Dakar urban database compiled by the Senegal flood prevention and management department (DPGI) and the Geographic Works and Cartography Department (DTGC). It is based on a fine resolution DTM (10m, resampled to 5m), from which natural drainage directions are extracted (Jenson and Domingue, 1988). These drainage directions are then forced to produce an associated drainage model that incorporates (i) the presence of buildings, (ii) collectors and (iii) retention basins. These operations were carried out using a combination of GIS tools and the Vicair module of the ATHYS modelling platform (http://www.athys-soft.org/), where specific algorithms were developed for this purpose (see Diémé et al. (2022) for more details).



### 3.2 Partitioning the study area into elementary basins and networks

The modified drainage model was first used to reconstitute elementary catchments with the same urbanised area. A threshold of 10 ha was adopted for this urbanised area.

The criteria for delimiting these small basins are taken from Jenson and Domingue (1988). They consist in marking the mesh as the outlet of a basin if:

- its drained area is greater than N (10ha)
- the difference in drained area between this mesh and the downstream mesh is greater than N.

In this way, 890 small urban basins were defined (Fig. 2a). The hydrographic network linking these small basins was defined as all the meshes draining at least an area equal to 1 ha (Fig. 2b). For this network, we differentiated between meshes with known geometric characteristics (widths and depths of the main canals, i.e. 297 sections) and those with unknown characteristics (either natural sections or sections with unknown dimensions). These meshes were given a different numerical code to apply a different parameterisation to the 1D kinematic wave model. In the drainage structure, this network is linked to a set of 106 retention basins. Each retention basin is reduced to a single mesh, representing its outlet. To this mesh is assigned a height-volume-drainage law to describe the operation of the reservoir and a unique identifier has been associated to each reservoir, so that their operation can be simulated differently (see section 4.3).

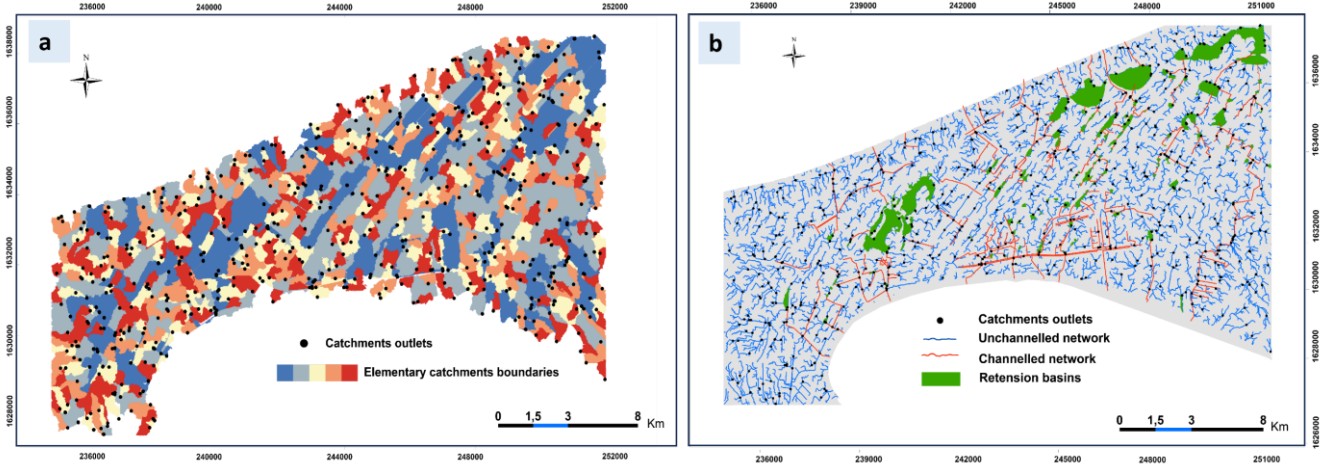

**Figure 2: a) Partitioning of the area into small urban catchment; b) Definition of runoff transfer and storage classes**

### 3.3 Application of hydrological runoff and routing model

#### 3.3.1 The SCS (Soil Conservation Service) runoff model

The SCS hydrological model (Ponce and Hawkins, 1996), which is often applied in small urban basins (Bouvier et al., 2018; Meng et al., 2019), was used to estimate the runoff on each mesh (Maref and Seddini, 2018; Bouadila et al., 2023). This model has the advantage not only of being relatively simple, capable of translating a trend towards an increase in the runoff coefficient as a function of rainfall, but also of being supplied with charts enabling S (or its CN equivalent) to be determined





as a function of: soil type and land use, land use density, initial moisture conditions (Steenhuis et al., 1995; Huang et al., 2007).

The model's adjustment parameters are Ia and S. The parameter Ia represents the initial losses before the onset of runoff (mm), and S the maximum water retention capacity of the soil at the start of the event (mm). The model is generally applied assuming that $Ia = 0,2.S$ , which is expressed by Eq. (1):

$$Q = \frac{(P-0,2.S)^2}{P+0,8.S} \qquad P > 0,2.S \ ; \ if \ not \ Q = 0 \tag{1}$$

where P is the total rainfall during the event (mm), Q is the runoff during the event (mm).

The dynamic formulation of this model (i.e. the temporal evolution of the flow during the event) is given (Eq. 2) by Gaume et al. (2004):

$$Pe(t) = Pb(t).\left(2 - \frac{(P(t)-0,2.S)}{P(t)+0,8.S}\right)\left(\frac{P(t)-0,2.S}{P(t)+0,8.S}\right) \tag{2}$$

Where Pe(t) represents the runoff produced (mm/h), Pb(t) the intensity of the rain received (mm/h), P(t) the cumulative rainfall since the start of the storm (mm). S is the only model adjustment parameter. The model is applied to each grid cell in

the area under consideration with a time step of 5 minutes, and S is likely to vary spatially depending on urban conditions.

### 3.3.2 The routing model

On each grid cell, the runoff provided by the SCS model is transferred to the outlet by the Lag and Route (LR) model (Fig. 3). Each mesh in the entire basin provides an elementary hydrograph, and the complete hydrograph is obtained at the outlet of each basin by summing the elementary hydrographs (Tramblay et al., 2011).

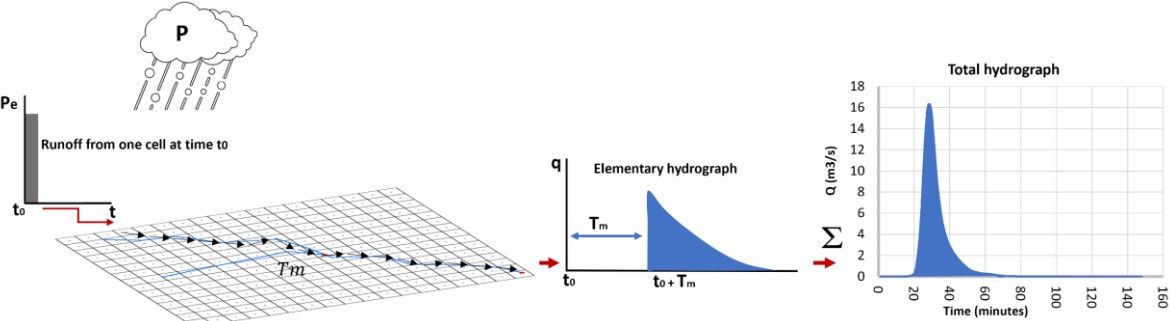

Figure 3: Conceptual diagram of the Lag and route model.

The elementary hydrograph provided by a mesh depends on 2 variables:

-   the transfer time Tm (Eq. 3) from the mesh to the basin outlet, equal to:




$$Tm = Lm/Vo \tag{3}$$

155        where Lm is the distance from the mesh to the outlet, Vo is the average velocity of the flow over the path travelled (possibly varying for each mesh)

-      the time Km (Eq. 4) associated with the diffusion of the flood wave during the transfer time Tm, equal to:

$$Km = Ko.\,Tm \tag{4}$$

where Ko is the proportionality coefficient between translation and diffusion (dimensionless).

Vo and Ko are the 2 parameters of the LR model.

The equation of the elementary hydrograph (Eq. 5) produced by the effective rainfall Pe(to) obtained on each mesh at each time to is given by:

$$Q(t) = \frac{Pe(t_0)}{K_m} \exp\left(\frac{t-(t_0+T_m)}{K_m}\right) A \qquad si\ t > t_0 + T_m \ \text{and}\ Q(t) = 0 \text{ if not} \tag{5}$$

where A is the mesh size (here $5m^2$). This LR model has the advantage of being based on fast calculation times, is
numerically stable and relatively easy to parameterise at the scale of the basin (Bouvier et al., 2017; Bouadila et al., 2023).

**3.4 The propagation model in the drainage network**

The velocity of flow propagation in the unchannelled network and the channelled network (297 collectors) is simulated by the Kinematic Wave (KW) hydraulic model (Vieux and Gauer, 1994) using the Manning-Strickler formula (Eq. 6), which
takes into account the characteristics (roughness, slope and cross-sectional dimensions) of each mesh of the network.
Its formula is given by:

$$V(t) = Kr.\sqrt{I}.\,Rh^{2/3} \tag{6}$$

where V(t) represents the flow velocity (m.s-1), Kr the Manning Strickler roughness coefficient ($m^{1/3}.s^{-1}$), I (m.m$^{-1}$) the slope of the land in the direction of flow, Rh (m) the hydraulic radius. The hydraulic radius is given by Eq. (7):
$$Rh = \frac{S_m}{P_m} \tag{7}$$

Where $S_m$ (m$^2$) denotes the wetted cross-section and $P_m$ (m) the wetted perimeter.

For a rectangular cross-section, these two parameters are obtained by Eq. (8) and Eq. (9):

$$S_m = H.\lambda \tag{8}$$
$$P_m = 2.H + \lambda \tag{9}$$





Where λ denotes the width of the section and H the height of water in the section. The OC model, being more physical (Singh and De Lima, 2018), thus requires longer computation times to calculate the mesh-to-mesh flows in the channelled and unchannelled network. The time saving is obtained by limiting the calculations to the network meshes, which means to a small number of meshes.

## 3.5 The water storage model in retention basins

A volume-height-discharge law, implemented in the Mercedes modelling module of ATHYS, provides information on the behaviour of each retention basin for different water levels. This law is tabulated and indicates, for different volumes stored in the retention basin, the corresponding height of water and the outlet flow from the reservoir.

Given the simple geometric shape and the type of drainage of the retention basins, this law can be summarised in 2 lines indicating the triplets heights-volumes-leakage rates for a zero volume in the reservoir (1st line) and for a maximum volume

in the reservoir (2nd line). Between the two lines, the heights will be linearly interpolated as a function of the volume stored in the reservoir.

## 4. Model parameterisation

Implementing the model on the scale of the entire study area requires parameterisation that takes into account: (i) soil types and urban conditions for the SCS model, (ii) the transfer rate of runoff for the LR model, (iii) the transfer rate in the network

for the 1D Kinematic Wave model, (iv) the storage capacity and discharge rate at the outlet of each reservoir for the storage model and (v) the project rainfall constructed to feed the models.

### 4.1 Parametrisation of the hydrological runoff and routing model

### 4.1.1 Parametrisation of the SCS runoff model

To calibrate the production model, we first measured rainfall and soil moisture on the very sandy soils often found in Dakar

(Diémé, 2023), and calculated infiltration by inverting soil moisture measurements (Le Bourgeois et al., 2016). Tests carried out using Hydrus 1D software (Šimůnek et al., 2016) showed that these soils were highly permeable, and capable of infiltrating rainfall in full. We then used hydrological data available for the city of Dakar (Fig. 4b) and measured in the Fann Mermoz experimental basin by Bassel et al. (1994) and Bassel and Pépin (1995). These data, collected during the two measurement campaigns, make it possible to evaluate the runoff coefficients for different rainfall events (Fig. 4b) in this

experimental basin, where the built-up coefficient was approximately 20% (Fig. 4a).



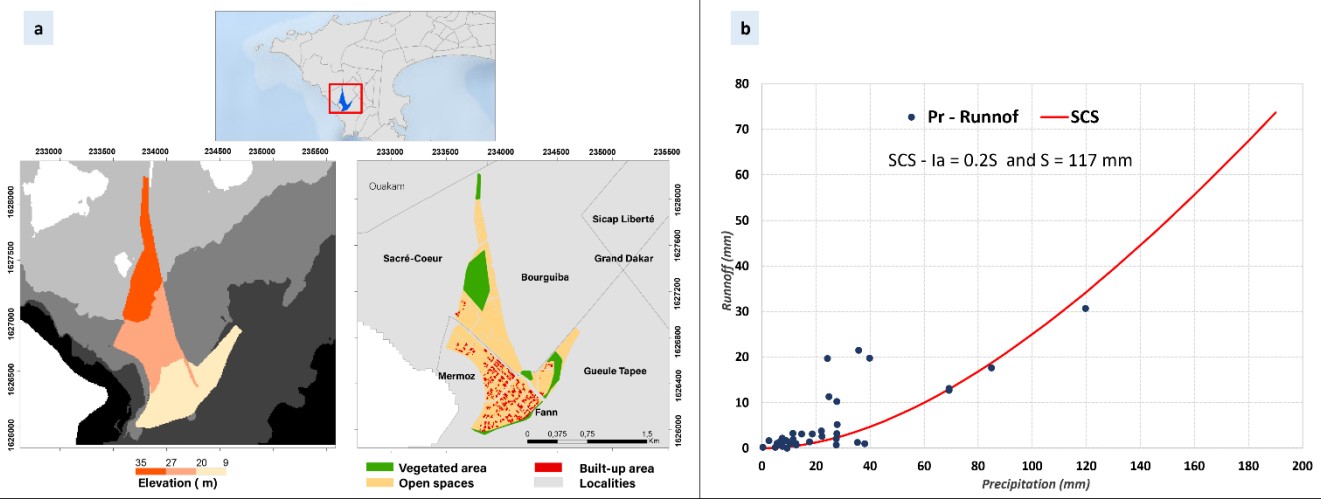

**Figure 4: a) Characteristics of the Fann Mermoz experimental basin; b) Relationship between rainfall and runoff (data taken from Bassel, 1996).**

Estimated runoff coefficients are of the order of 10% for a rainfall of 40 mm, 20% for a rainfall of 78 mm and 30% for a
rainfall of 150 mm. In other words, the basin's build coefficient is close to the runoff coefficient associated with a rainfall of 78 mm, i.e a rainfall with a ten-year return period in Dakar (Sane et al., 2018). This is consistent with the filtering nature of the soil, as we have characterised, which indicates that unpaved soils produce negligible runoff for most rainfall events. To obtain a runoff coefficient of 20% with a rainfall of 78mm, the value of the S parameter of the SCS model must be set to 117mm.

Finally, we generalised the assumption that the building coefficient is equal to the runoff coefficient associated with a ten-year rainfall for the entire Dakar site. The building coefficients were calculated for each urban block, as the ratio of the surface area of the buildings to the surface area of the block (Fig. 5). All the meshes within the same urban block were then assigned the value calculated for the block.



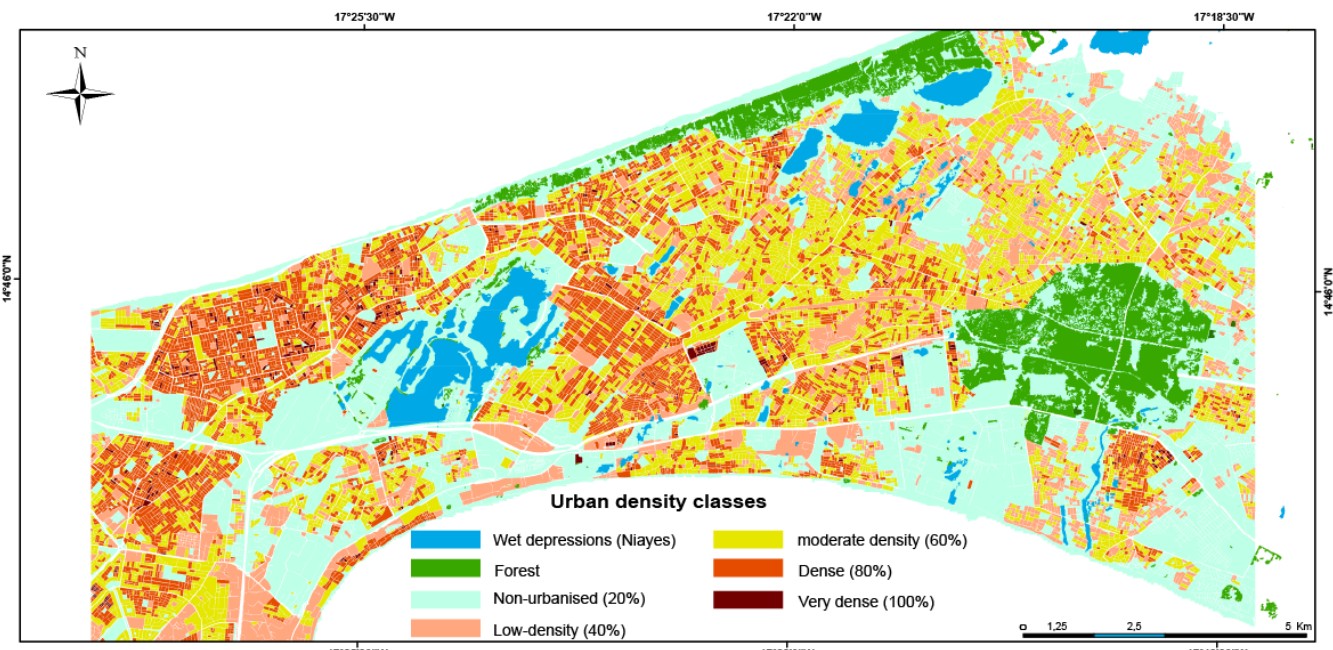

**Figure 5: Determination of urban density classes by urban block**

Finally, we calculated the values of S for different classes of building coefficient, following Eq. (10):

$$CR = \frac{Q}{P} = \frac{(P-0.2S)^2}{P.(P+0.8S)} \tag{10}$$

Where P is the height of the 10-year rainfall in 4 hours, i.e. 78 mm. The values obtained after adjusting the S parameter are shown in Table 1.

**Table 1: Summary of values obtained for the S parameter of the SCS model**

| Urban density classes % | Run-off coefficient (ten-yearly) | S values |
|:---:|:---:|:---:|
| 20 | 20 | 117 |
| 40 | 40 | 67 |
| 60 | 60 | 35 |
| 80 | 80 | 15 |
| 100 | 100 | 0 |

**4.1.2 Parametrisation of the routing model**

The Ko parameter was set at 0.7, the empirical value usually used (Bouvier et al., 2017). The Vo parameter was determined using historical data from the Fann Mermoz experimental basin (Bassel, 1996).





In this study, we considered the basin response time (Tr) to be the same as that provided by the LR model, applied to the
mesh occupying the center of gravity of the basin's active zone (urbanized area). This mesh is located approximately 1.2 km
from the basin outlet. The theoretical response time provided by the model is given by Eq. (11):

$$Tr \; = \; Tm \; + \; Km \tag{11}$$

With m the mesh of the basin's center of gravity, this leads to Eq. (12):

$$Tr \; = \; 1.7 \, Tm = 1.7 \, Lm/Vo \tag{12}$$

And therefore Vo is obtained using Eq. (13):

$$Vo \; = \; 1.7 \, Lm/Tr \tag{13}$$

Response times were estimated at an average of 30 minutes based on available rainfall-runoff observations, taking into
account the difference between peak rainfall and peak runoff. For a distance from the center of the basin to the basin outlet
equal to 1.2 km, the calculated flow transfer velocity (Vo) is equal to 1.1 m/s.

This parameterization based on data from the experimental basin was then extrapolated to the scale of the study area,
considering the transfer rate to be uniform and equal to that obtained for the Fann-Mermoz basin for all rainfall events. This
approximation is justified by the fact that slopes vary little in Dakar, and are on average fairly close to those of the Fann-
Mermoz basin.

## 4.2 1D hydraulic model parametrisation

### 4.2.1 Propagation in the unchannelled network

The unchannelled network meshes here have been linked to both (i) the right-of-way of streets and roads which, in urbanized
areas, become the transfer pathways for surface runoff (Zhang et al., 2018; Skrede et al., 2020) due to the presence of
buildings, walls and other urban objects (Fig. 6) that divert flows (Diémé et al., 2022) and (ii) the shallow natural reaches
arising from non-urbanized surfaces. The directions of flow that can pass through these unchannelled meshes were defined
when constructing the drainage topology (section 3.1).

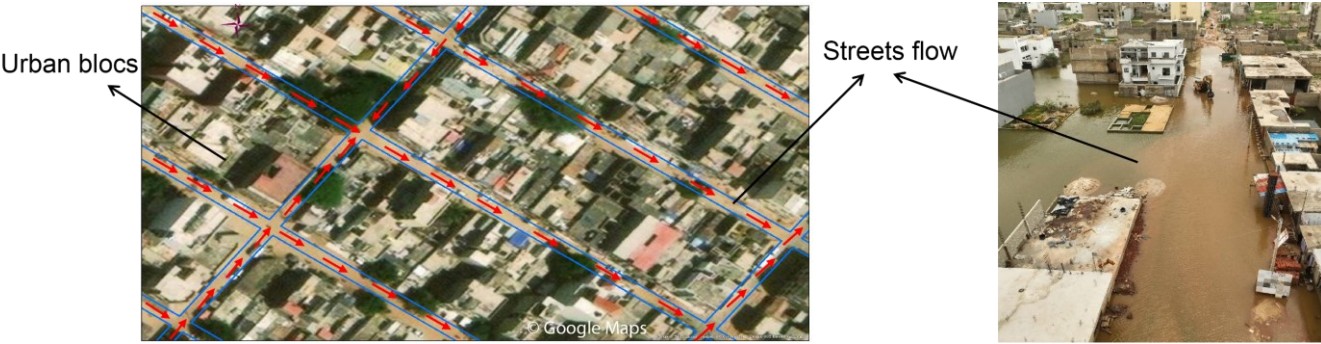

**Figure 6: Surface water drainage paths at urban street level.**



The network meshes derived from streets and roads were classified into types (residential streets, primary-secondary roads,
national roads, freeways), and assigned a specific width according to their spatial footprint. This was done interactively, on-
screen, by superimposing the city's roads and streets layer onto a satellite imagery background (Google Earth). The widths
corresponding to each category of road were estimated on a case-by-case basis, taking into account the alignment of the
carriageway and its verges. The Figure 7 shows the values found (in meters) and defined using Google Earth's distance
measurement tools. As there are no constraints (walls, partitions, etc.) in the propagation of natural reaches, they have been
considered in the model as having infinite width.

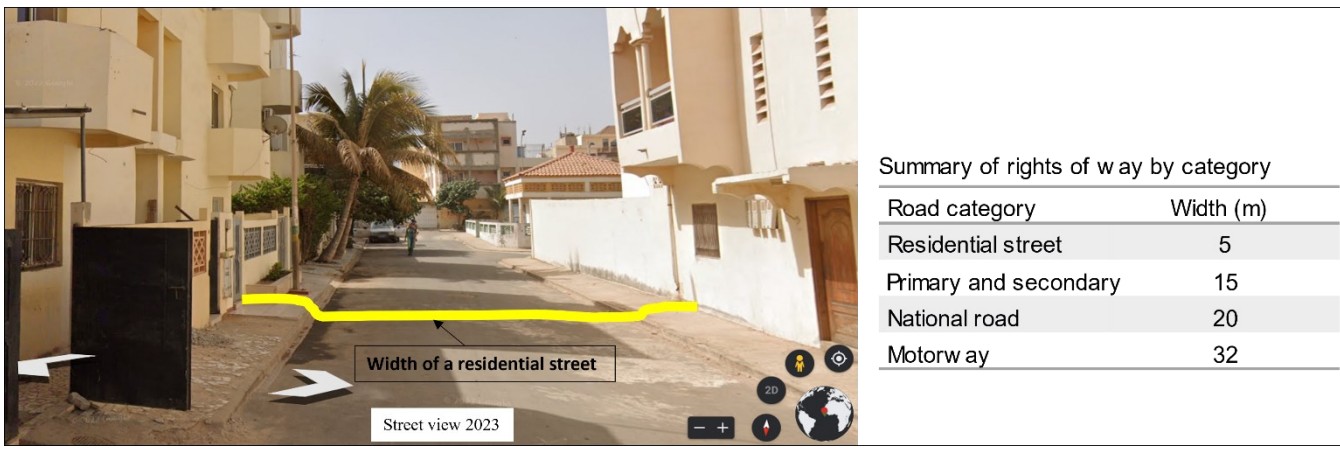

**Figure 7: Determining the widths of unchannelled transfer classes**

With regard to the depth of the minor bed (Pc) associated with street and road meshes, an infinite depth has been set, so that
the flow remains channelled by the width of the road or street (in this case by the walls bordering the street) and considered
to be null for the meshes of the natural reaches, which are very little marked in Dakar. Manning Strickler roughness
coefficients were estimated at 50 $m^{1/3}.s^{-1}$ for all street and road meshes with more or less smooth surfaces, and at 20 $m^{1/3}.s^{-1}$
for natural meshes with rough surface.

Slope values were calculated from the DTM, using the differences in elevation at the nodes of each mesh, in the mesh's
drainage direction, then smoothed, in order to limit the sensitivity of the 1D Kinematic Wave model to slope variability
(sometimes linked to DTM accuracy shortcomings). Smoothing was based on the difference between the altitudes of the
mesh and the $N^{th}$ mesh downstream, divided by the length of the trajectory between the mesh and the $N^{th}$ mesh downstream.
The number N of meshes used for smoothing has been set at 50 meshes. If the calculated slope is equal to 0 (or even <0) or
there is no $N^{th}$ mesh (on the edges of the image), this slope is assigned the value 0.001 m/m. Slopes smoothed in this way
range from 0.001 to 0.4 m/m, with an average of 0.007 m/m.



### 4.2.2 Parametrisation of the propagation model in the channelled network

The parameters λ and Pc were set on the basis of the dimensions of the collectors for which information on their characteristics (width and depth) is available in the preliminary and detailed technical reports produced as part of PROGEP. The roughness coefficient has been uniformly set at 50 $m^{1/3}s$, and flows are calculated over all channel sections, taking into account the overall rectangular cross-section. The slopes applied are obtained by smoothing the DTM altitudes.

### 4.3 Parametrisation of the water storage model

In the list of retention basins (106), only the retention basins built as part of the PROGEP project (84 basins) have detailed information (storage capacity, height and discharge rate), which we have extracted from the various reports produced by this project. As for the other basins whose dimensions are not known, we have assigned them, by default, characteristics based on a criterion of similarity of the shapes of their contours with those of basins whose dimensions are known, and by visually comparing them using satellite imagery from Google Earth. The 106 selected have been included in the flow modeling chain, and their operation at different water levels is simulated by a volume-height-flow law defined in the ATHYS Mercedes module. In the simulations, leakage flow rate has a constant value (Table 2), corresponding to the capacity of the nozzle or the drainage channel located at the reservoir outlet, at the lowest level of the reservoir bottom.

**Table 2: Example of a reservoir operation**

| Height (m) | Volume ($m^3$) | Emptying rate ($m^3$/s) |
|:---:|:---:|:---:|
| 0 | 0 | 3,90 |
| 1,2 | 15 000 | 3,90 |

When maximum reservoir capacity is reached, the flow entering the reservoir is fully transferred downstream. However, when there is no outlet channel, the entire volume of water is stored in the reservoir, so there is no transfer downstream.

### 4.4 Project rainfall construction

Project rainfall was constructed to be injected into the model and simulate runoff discharges (Zhenyu and Olivier, 2005; Balbastre-Soldevila et al., 2019). They were constructed from the IDF curves that we calculated using the GEV law parameter values (μ, σ, ε) established by Sane et al. (2018), for each region of Senegal. In order to obtain a reliable estimate of the distribution parameters μ, σ and ε for each rainfall station, Sane et al. (2018) mixed all the data of different durations considering that the distributions of rainfall of duration d are identical to within one factor, η, called the scaling factor. This approach makes it possible to create a single sample of rainfall of different durations by grouping together all the maximum annual rainfall values of all durations d, after dividing each rainfall of duration d by a quantity $d^\eta$. Finally, they fitted a GEV





distribution to this sample, with the Dakar parameters μ = 28.9 mm, σ = 12.5 mm, ε = 0.08, with η set to η = -0.86. The parameters associated with the rainfall distributions for each duration *d* are obtained using Eq. (14), Eq. (15) and Eq. (16):

$$\mu(d) = \mu.d^{\eta} \tag{14}$$

$$\sigma(d) = \sigma.d^{\eta} \tag{15}$$

$$\varepsilon(d) = \varepsilon \tag{16}$$

This makes it possible to determine the maximum rainfall of duration *d* (1 to 24 hours) corresponding to each return period of 2 years to 100 years (Fig. 8a).

This maximum rainfall, calculated for different return periods, was then used to construct the project rainfall. The design rainfall model used is the double triangular rainfall model proposed in France by Desbordes and Raous (1976). This form of

synthetic hyetogram, in which the position of the rainfall intensity peak is centred, has the advantage of guaranteeing hydrological models' maximum efficiency in calculating hydrographs (Roux et al., 1995). It takes into account the total duration of the rain, t3, whose height taken from the IDF is equal to P(t3,T), the period of intense rain of duration t1, whose height P(t1,T), is also taken from the IDF and a period t2 which constitutes a period of rain before and after the intense period. For its construction, the basic parameters to be determined are $i_m$ (the maximum intensity before the intense period)

and iM (the maximum intensity of the peak of the intense rainfall). The $i_m$ is calculated following Eq. (17):

$$i_m = \frac{P(t3,T)-P(t1,T)}{t2} \tag{17}$$

Where P is the height of the rainfall, T is the return period, t3 is the total duration of the rainfall with a period fixed here at 4 hours, representative of the average duration of rainfall in the region, t1 is the duration of the intense rainfall, and corresponds to the time of concentration in the basin (fixed here at 1 hour due to the nature of the data from the IDF curves,

which do not provide sub-hourly intensities).

The second parameter iM is calculated following Eq. (18):

$$iM = \frac{2P(t1,T)}{t1} - i_m \tag{18}$$

Based on $i_m$ and iM, rainfall intensities are then determined for every 5 minutes by linear interpolation between times 0 and t2. ; $t2 \; and \; t2 + \frac{t1}{2}; t2 + \frac{t1}{2} \; and \; t2 + t1; t2 + t1 \; and \; t3$. The project rainfalls constructed with t1 = 1h and t3 = 4h for

different return periods are shown in Figure 8b.





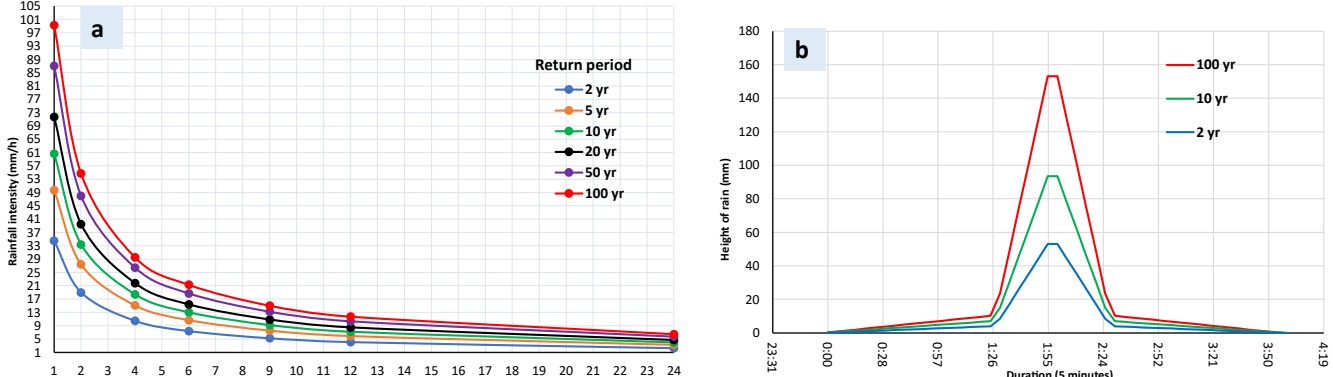

**Figure 8: (a) IDF rainfall curves for Dakar calculated from the GEV parameters defined by Sané et al. (2018); (b) Construction of the project rainfall for return periods of 2, 10 and 100 years with an intense period of 1 hour.**

## 5. Results and discussion

### 5.1 Application to detecting network overflows

Running the simulation model provides the flows, heights and velocities in the drainage and storage networks over the course of the event. The overflows from the network correspond to a positive difference between the simulated flows and the full-load capacities for the drainage network, and to a positive difference between the simulated volume and the maximum volume for the storage basins. The capacities of the drainage network were calculated by applying the Manning-Strickler formula at full load, i.e. with a head of water corresponding to the depth of the structure. The results are maps of overflows from the stormwater drainage network and retention basins at the scale of the study area. The results presented here are based on running the model with 10-year (78 mm; Fig. 9) and 100-year (128 mm; Fig. 10) project rainfall over a 4-hour period. The simulations were carried out on the assumption that the rainfall was uniform over all the defined catchment areas.

The simulations carried out show that the network records significant overflows for rainfall with a 10-years return period, with overflow levels ranging from 1-12 $m^3$/s for some sections and 12-18 $m^3$/s for three sections. For most of the collectors that overflowed, the overflow rate varied between 1 and 6 $m^3$/s.



**Figure 9: Identification of network overflow points for a 10-year return period rainfall.**

Above the ten-year rainfall frequency, the network shows widespread overflow levels. This applies to both collectors and

storage basins. However, some reservoirs are still operational. The large natural depressions (the Niayes) do not overflow.

Overflows are noted on a large proportion of the collectors, with thresholds sometimes exceeding 18 m³/s in some cases. The

simulations also show that the flow load has caused a large number of retention basins to overflow.





**Figure 10: Identification of network overflow points for a 100-year return period rainfall.**

## 5.2 Discussion

The model used in this study has the advantage of being relatively simple and is capable of covering an entire city with a fine resolution ($5m^2$) and short calculation times (typically 5 minutes). This makes it a useful tool for assessing flood risk. The model is also compatible with real-time flood forecasting applications, if remote rainfall data is available. However, this



study has a number of limitations. An important factor is the construction of the drainage topology based on the DTM, which has focused solely on the effects of the location of buildings, canals and storage basins in modifying drainage directions. To obtain a more detailed view, the analysis should be extended to include other urban objects that influence the trajectories of surface runoff flows. Future work will focus on lidar data (currently being compiled for the Dakar region), which provides more detail than DTM on urban micro-objects and could thus be used to refine the reconstruction of

induced drainage directions. The other limitation of this work relates to the availability of the data (hydrological, hydrometric or piezometric) required to parameterise the hydrological runoff-routing models (SCS-LR) applied to this drainage topology in order to calculate flows. The parameterisation of the SCS runoff model was based on the hypothesis that we established, using short series of data from the Fann Mermoz experimental basin, by considering that the ten-year runoff coefficient is equal to the building coefficient. Although this simplification may be acceptable for the case of Dakar,

where soils are generally sandy and very permeable (Diémé, 2023), new data must be produced to verify the hypothesis. In other cities where the soils are less permeable, direct (Kelleners et al., 2005) or indirect (Galagedara et al., 2003) infiltration measurements on several representative sites should be used as a basis for determining the contribution of these soils to surface runoff. The LR routing model was calibrated by considering the transfer velocity (Vo), calculated on the Fann Mermoz experimental basin, as uniform over the entire study area. The slope conditions, which vary little in the study area,

allow us to retain this approximation. It is clear that the parameterisation of both the SCS-LR runoff-routing models needs to be improved using new experimental data, which is relatively rare in African cities. This should motivate the setting up of new experimental sites, in order to better estimate the parameters of the flow calculation models.

One of the ways in which the model has been improved is in the calculation of the slopes used by the OC hydraulic model to ensure the propagation of flows in the channelled and unchannelled network. A simplification has been applied, involving

smoothing to reduce the sensitivity of the OC model to irregular variations in the slope of the terrain, sometimes linked to a mistake in the DTM. The availability of lidar data in the study area will enable us to compare the model's performance using more accurate slopes. Similarly, the congestion of collectors (household waste, silting, etc.) can be incorporated into the hydraulic model. This could be taken into account by reducing the cross-section of the collector, even if information on this congestion is difficult to obtain. A particular constraint is the influence of the rising water table in Dakar (Faye et al., 2019),

which must be taken into account in the hydrological production model. One possible solution is to obtain piezometric data giving the water table level and reduce the S parameter of the SCS model on the meshes corresponding to outcrops of the water table. Also, to take better account of the extreme precipitation regime, it would be interesting, instead of stationary DFIs (used in this study), to explore non-stationary statistical methods for determining DFIs (Chagnaud et al., 2021) that incorporate the uncertainties associated with climate change. Finally, validating the results of the model simulations is one of

the major perspectives of this study. This could be done by using information on feedback, recent flooding situations and data on the intensity of rainfall events that cause flooding. An identical approach has already been proposed for the city of Bamako (Mali) by (Chahinian et al., 2023). The aim is to compare simulations of flood situations that have already occurred

with flood maps or feedback data. Validation of the method would enable it to be extended to other towns and cities, thereby ensuring sound planning decisions.

**6 Conclusions**

A fine-scale simulation model of runoff over an entire urban area and an assessment of the response of the storm drainage network (canals and retention basins) to different rainfall events has been developed. It is based on a preliminary reconstruction of the drainage directions modified by urbanisation and the implementation of combined hydrological and 1D hydraulic models calibrated to the city's urban conditions.

The results obtained are overflow maps for the city's drainage network for rainfall intensities of different return periods. The representation of overflow points is associated here with one-dimensional modelling, but is still sufficiently informative to guide the deployment of emergency services on the ground, or to initiate action at strategic locations: assessment of the effectiveness of planned developments, tests of different rainfall and urbanisation scenarios, detection of overflows in near-real time with remote rainfall data. In addition, the model also provides boundary conditions for applying 2D hydraulic

models to determine locally the propagation of overflows from stormwater drainage network over limited areas. Future work will focus on improving the availability of data to facilitate the assessment of simulation uncertainties and validate the overflow results. Indeed, one of the challenges of urban hydrology in African cities is to set up urban databases that are essential for conducting relevant studies and for better characterising and forecasting floods.

**Data Availability**

The data used in this article are available from the first author, LPM Diémé, upon reasonable request.

**Author contribution**

LPMD has conceived and designed the analysis. CB has implemented the tools codes on ATHYS. AS has provided the data and AB has contributed to the data analysis

**Competing interests**

The authors report no conflicts of interest.



**Acknowledgements**

This article was produced with the support of the Water Cycle and Climate Change (CECC) 2021-2025 project, co-funded by IRD and AFD. The authors would also like to thank the technical structures that agreed to make the data used in this
article available.

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
