# Peer review of "Modelling urban stormwater drainage overflows for assessing flood hazards: Application to the urban area of Dakar (Senegal)"

_EGUsphere, 2023_

## Referee Comment (RC1)

**egusphere-2023-2458 (NHESS)**

Title:  **Detection of flooding by overflows of the drainage network: Application to the urban area of Dakar (Senegal)**

Authors :  Laurent Pascal Malang Diémé et al.

**GENERAL COMMENTS**
The paper presents the development of a simple modelling system for the detection of urban drainage network overflow.  The drainage system consists of artificial drainage and storage basins. The model has been developed and applied on the urban outskirts of Dakar, Senegal. It aims to propose a method capable of modeling flows at fine resolution (5m2), over the entire area, and providing a rapid diagnosis of how the drainage network is operating for design rainfall intensities The steps followed to the development of the modelling system are: i) determination of drainage directions, ii) application of a hydrological model for estimating flows at the outlets of elementary catchments, iii) the implementation of a hydraulic model for propagating these flows through the drainage network and iv) application of a simple storage model for the simulation of retention basins. The network overflow points are calculated as the difference between the calculated flows and the network's capacity to evacuate them.

My major and minor comments are presented in the next paragraphs.

**Major Comments**
There are many points that should be clarified before considering the paper for publication.

1.  A flow chart of the methodology should be used to present the methodology.   This will help the reader to understand the proposed modelling system.
2. It is not clear to me whether the drainage system (i.e. stormwater drainage network and retention basins) is constructed or it is planned. It is strange to me that the drainage network is a network of open channels of orthogonal cross section. Stormwater drainage network is usually underground and consists of pipes.  If the network exists then the dimensions are set and known otherwise the dimensions of the drainage network elements (i.e. pipes and cannals) is a matter of design.  The authors should clarify this issue.
3. In continuation of the previous comment, more information about the study area should be presented, e.g. climate, historical extreme rainfall and flood events, hydrology, DEM, etc.
4. More information about the Kinematic Wave (KW) flow routing model should be given in Section 3.4.  The governing equations of KW to be solved should be presented.
5. Section 3.5.  Why a simple linear storage model is not used for water retention structures?
6. All areas have the same soil characteristics as found in the experimental site.  It would be more realistic to have a soil map of the area or CN maps to estimate the

parameters of SCS rainfall abstractions (or effective rainfall) model.

7. What is the basin response time (Tr)? Is it the time of concentration or the time lag? In Equation 11, please explain what is Tm (transfer time). Why Tr is not estimated by widely used common and typical equations?

8. Give the general equation of IDF curves as $i = CT^n D^{-k}$

9. How and why a 4-hour rainfall is selected? Is 4 hours the critical duration of a storm? Please explain.

10. Why the spatial distribution of design rainfall is not considered? The same design hyetograph is applied over the study area.

11. A major drawback of the study is that the methodology has not been validated against historical flood events. The results presented are purely theoretical and could be fictional and not representative. The authors should simulate one or two events for validating the method and the modelling system.

12. Conclusions. The authors correctly write the deficiencies of the methodology but they should outlined and discussed these deficiencies earlier in the paper.

13. There are many awkward hydrological terms. Proper hydrological terms should be used. Some of them are indicated in the minor comments bellow.

14. English language needs improvement. In some paragraphs, the English writing is poor.

**Minor comments**
1. There many improper hydrological terms. For example:
   a. Line 93. "…hydrological production …." Please revise to "…..flow generation…."
   b. Line 97. "……injected in the model….." Please revise to "……used as input data to the model…"
   c. Line 199. "…production model…" Please revise to "….hydrological model…."
   d. Line 308. "…..project…" Use the term "design"
      And others.

2. Equation 5. Not "si". It is "if"
3. Line 180 and elsewhere. What is the OC model? It has not been described.
4. Table 2. It is not understandable. Use the equation of reservoir level-storage volume-outflow curves.

The presented study falls within the scope of NHESS. However, the paper is not ready for publication and needs at least *major* revisions before it would be acceptable for publication in the journal of NHESS.

---

## Author Comment (AC1)

**Response to the Reviewers Comments**

**Author reply to RC2 egusphere-2023-2458 (NHESS)**

**Paper title:** Detection of flooding by overflows of the drainage network: Application to the urban area of Dakar (Senegal)

Authors : Laurent Pascal Malang Diémé, Christophe Bouvier, Ansoumana Bodian and Alpha Sidibé

We sincerely thank the reviewer for providing these comments on this manuscript. Below, we address all of these comments. The reviewer comments are displayed in blue, while the author response are displayed in black colour.

**Reviewer's report: Referee #2**

**General comments**

This study aims to model a fine-scale run-off model of the urban area and assessment of the response of the storm drainage network (canals and retention basins) to different rainfall events. The methodological approach is based on a preliminary reconstruction of the drainage directions modified by urbanization and the implementation of combined hydrological and 1D hydraulic models calibrated to the city's urban conditions. My minor and major comments for this manuscript are as follows.

1. **Minor comments**

Grammatical errors

Line 42 – 44: There is a repeated sentence here.

The sentence will be deleted

Figure 1. Please insert the north arrow into the map

The north arrow was inserted into the map in Figure 1. An altitude map and a soil types map will be also added to provide a better description of the study area.

[Figure]

Figure: (a) Location of the study area; (b) Digital Terrain Model; (c) soil type distribution

**2. Major comments**

- The title does not match the main objectives of this study. The authors should confirm the aim of this study focusing on the detection of floods or modelling of drainage networks.

We propose to change the title "Detection of flooding by overflows of the drainage network: Application to the urban area of Dakar (Senegal)" to "**Modelling urban stormwater drainage overflows for assessing flood hazards: Application to the urban area of Dakar (Senegal)**"

- Please insert a flow chart of data processing

A flow chart of the method will be inserted in section 3.

[Figure]

- The study should integrate the validation of flood simulation results using accurate reference data.

Indeed, a key limitation of this study is the lack of validation of the simulation results, as it was pointed out in the discussion. Although it is of major importance, the validation task cannot be undertaken at the moment. However, we hope that the methodological aspects of the simulation should be of enough interest to be published. The validation task is a priority and a perspective for our future work. We have added the following sentence to the discussion section:

"As things stand at present, it was not possible to get the necessary data for the validation of the method, which means on the one hand sub-daily rainfall data, and on the other hand flood maps for the recent events that occurred in Dakar.  The imminent installation of a rain gauge radar in Dakar, as part of the integrated flood management project in Senegal, could help to facilitate this.  Flood maps could be obtained by exploring citizen science tools (Sy et al., 2020) or ordering a high-precision satellite image to map out flooded areas."

- Section 3.1. please give a brief explanation about ATHYS modelling that was applied in your study

In addition to the internet link to the ATHYS platform, a brief description of ATHYS has been added at the beginning of section 3. The sentence is as follows:

"The modelling chain was built in the ATHYS platform, developed by Hydrosciences Montpellier. ATHYS enables a range of hydrological and hydraulic GIS-based models, as well as geographical (VICAIR) and hydrometeorological (VISHYR) data processors.  ATHYS is a free software, available from www.athys-soft.org."

- What is the source of the DTM data and soil map used in this study? Please explain in detail about the resolution as well as the accuracy of these data.

Source and details of the DTM and soil map will be added in section 2.

- Line 155: how the model calculates the average velocity of the flow (Vo) is not specified.

The parameter Vo (velocity of the surface flow) was determined using data from the Fann-Mermoz gauged catchment. These data allow us to set a fixed velocity of 1.1 m/s (Lines 235-239). The velocity Vo is used to determine the transfer time Tm (Eq. 3) integrated in the model (Eq. 5).

- Section 5 – Results and discussions part also does not match the main objectives of this study. Section 5.1 should robustly specify the results of flood simulation using the ATHYS modeling method as three methodological steps shown in the abstract.

The discussion section will be revised. Additional details on the identified method validation strategies will be included.

---

## Author Comment (AC2)

**Response to the Reviewers Comments**

Author reply to RC1 egusphere-2023-2458 (NHESS)

Paper title: Detection of flooding by overflows of the drainage network: Application to the urban area of Dakar (Senegal)

Authors : Laurent Pascal Malang Diémé, Christophe Bouvier, Ansoumana Bodian and Alpha Sidibé

**General comments**

The presented study falls within the scope of NHESS. However, There are many points that should be clarified before considering the paper for publication.

We thank the reviewer for the valuable comments and appreciate the useful suggestions to improve the manuscript. In the revised manuscript, we will carefully consider the reviewer comments. Below, the reviewer comments are displayed in blue, while the author responses are shown in black. We expect these changes have improved the readability of the text and its structure.

**Major Comments**

**1.** A flow chart of the methodology should be used to present the methodology. This will help the reader to understand the proposed modeling system

A flow chart will be inserted in section 3.

[Figure]

**2.** It is not clear to me whether the drainage system (i.e. stormwater drainage network and retention basins) is constructed or it is planned. It is strange to me that the drainage network is a network of open channels of orthogonal cross section. Stormwater drainage network is usually underground and consists of pipes. If the network exists then the dimensions are set and known otherwise the dimensions of the drainage network elements (i.e. pipes and cannals) is a matter of design. The authors should clarify this issue

A more detailed description of the stormwater channel structure, retention basins, and DTM will be provided in section 2. In the study area, the drainage and stormwater storage network already exists (lines 85-87). The drainage network built by the stormwater management project (PROGEP) and its dimensions are provided in numerous technical reports of the project. In Dakar, for cost and facility of maintenance, the entire rainwater network is structured as an open-air network (canals, ditches) or

sometimes an underground network (pipes, gutters). The majority of channels are surface drains and are rectangular.

**3.** In continuation of the previous comment, more information about the study area should be presented, e.g. climate, historical extreme rainfall and flood events, hydrology, DEM, etc.

The description of the study area will be extended to include relief, climate, and the various flood situations that have affected the city. Also, a figure showing the distribution of elevations, soil types will be added to provide a better description of the study area.

**4.** More information about the Kinematic Wave (KW) flow routing model should be given in Section 3.4. The governing equations of KW to be solved should be presented.

The 1D kinematic wave model will be more carefully presented in section 3.4. Below is a description :

"The hydraulic propagation velocity of flow in the unchanneled network and the channeled network (297 collectors) is computed by the 1D Kinematic Wave (KW) model (Constantindes, 1981; Miller, 1984). The KW model combines a conservation equation (Eq.):

$$\frac{\partial A}{\partial t} + \frac{\partial Q}{\partial x} = 0$$

where Q is the flow rate (m³/s), A is the area of the wetted section (in m²), $x$ is the horizontal distance (m) and t is the time (s).

with a dynamic equation, used as the Manning-Strickler formula (Eq. ) :

$$V(t) = Kr.\sqrt{S_f}.Rh^{2/3}$$

where V(t) represents the flow velocity (m.s⁻¹), Kr the Manning Strickler roughness coefficient (m$^{1/3}$.s⁻¹), $S_f$ (m.m⁻¹) the friction slope, Rh (m) the hydraulic radius, using :

$$S_0 = S_f$$

where $S_0$ is the bed slope (m.m⁻¹) .

**5.** Section 3.5. Why a simple linear storage model is not used for water retention structures?

We will modify the behavior of the reservoirs as suggested by the reviewer. Adopting a linear reservoir behavior is indeed more convenient. It was done by considering that the reservoir discharge was linearly dependent on the volume in the reservoir. So section 4.3 will therefore be rewritten in that sense, and the simulations will be redone under this new basis.

**6.** All areas have the same soil characteristics as found in the experimental site. It would be more realistic to have a soil map of the area or CN maps to estimate the parameters of SCS rainfall abstractions (or effective rainfall) model.

A soil map, provided by the National Institute of Pedology (Senegal), will be inserted and commented on in section 2.

**7.** What is the basin response time (Tr)? Is it the time of concentration or the time lag? In Equation 11, please explain what is Tm (transfer time). Why Tr is not estimated by widely used common and typical equations?

Tr is indeed a lag time, considered as the time between the center of gravity of the runoff and the center of gravity of the rainfall. The measurements of both rainfall and runoff in the experimental catchment of Fann-Mermoz led to an estimation Tr = 30 mn. As Tr can be also derived from the model under some simplified hypothesis, it allows to estimate Vo (the average velocity of the flow over the path travelled). This seems to be more reliable than empirical common equations. Section 4.1.2 will be rewritten to give more detailed explanation of the method.

**8.** Give the general equation of IDF curves as $i = CT^n D^{-k}$

We're going to insert a new table showing rainfall of different durations (1, 2, 4, 6, 9, 12, 24h) and different return period (10, 100 years).

**9.** How and why a 4-hour rainfall is selected? Is 4 hours the critical duration of a storm? Please explain.

4 hours is mostly the life duration of rainstorms generally observed in African convective systems as analysed by Tadesse and Anagnostou (2010). We have applied this duration for Dakar, even though detailed local analyses are required to assess the structure of significant precipitation events. The former critical duration of rain that we extracted from the IDF provided by Sane et al. (2018) is 1 hour. For future work, possible improvements would be to choose the critical duration less than 1 hour. Integrating durations less than 1 hour has been tested using the IDF curves recently updated by Diedhiou et al. (2024) . New simulations with the model show that the project rainfall associated with a critical duration of 1 hour tends to underestimate peak flows by an average of 7% compared with the project rainfall associated with a critical duration of 30 mn, regardless of the surface area of the basins, ranging from 10 ha to 12 km². Furthermore, a 10 mn led to underestimate the peak flows associated to a 30 mn critical period by an average of 9%.

[Figure]

*Comparison of peak flow between an intense rainfall of 30 minutes and 1 hour duration.*

**10.** Why the spatial distribution of design rainfall is not considered? The same design hyetograph is applied over the study area.

The simulations were carried out on the assumption that the rainfall was uniform over all the defined catchment areas. Such hypothesis does not account for areal reduction factor, but can be adopted

because most of the catchments (i.e. 94%) have small areas, less than 2 km². The largest catchment areas (6 to 12km²) account for 2% and 4% out of the 890 catchment outlets which have been considered, as shown below.

[Figure]

**11.** A major drawback of the study is that the methodology has not been validated against historical flood events. The results presented are purely theoretical and could be fictional and not representative. The authors should simulate one or two events for validating the method and the modelling system.

Indeed, a key limitation of this study is the lack of validation of the simulation results, as it was pointed out in the discussion. Although it is of major importance, the validation task cannot be undertaken at the moment. However, we hope that the methodological aspects of the simulation should be of enough interest to be published. The validation task is a priority and a perspective for our future work. We added in the discussion the sentence, as follows:

"As things stand at present, it was not possible to get the necessary data for the validation of the method, which means on the one hand sub-daily rainfall data, and on the other hand flood maps for the recent events that occurred in Dakar. The imminent installation of a rain gauge radar in Dakar could help to facilitate this. Flood maps could be obtained by exploring citizen science tools (Sy et al., 2020) or ordering a high-precision satellite image to map out flooded areas."

**12.** Conclusions. The authors correctly write the deficiencies of the methodology but they should outlined and discussed these deficiencies earlier in the paper.

A mention of one of the limitations of the work (in particular the non-validation of the simulations) will be included in the abstract.

**13.** There are many awkward hydrological terms. Proper hydrological terms should be used. Some of them are indicated in the minor comments bellow.

Corrected

**14.** English language needs improvement. In some paragraphs, the English writing is poor.

We will carefully correct this aspect

**Minor comments**

*1. There many improper hydrological terms. For example:*
*a. Line 93. "…hydrological production …." Please revise to "…..flow generation…."*
*b. Line 97. "……injected in the model….." Please revise to "……used as input data to the model…"*
*c. Line 199. "…production model…" Please revise to "….hydrological model…."*
*d. Line 308. "…..project…" Use the term "design"*
*And others.*

We have reviewed the terms and corrected them accordingly. We have also removed words or phrases that were repeated (lines 43-45, 51)

*2. Equation 5. Not "si". It is "if"*

Corrected

*3. Line 180 and elsewhere. What is the OC model? It has not been described.*

OC has been changed to KW which is the used kinematic wave model

*4. Table 2. It is not understandable. Use the equation of reservoir level-storage volume outflow curves.*

Table 2 will be modified according the use of the linear reservoir model

**References**

Constantindes, C. A.: Numerical techniques for a two-dimensional kinematic overland flow model., Water SA, 7, 234–248, 1981.

Diedhiou, C. W., Panthou, G., Diatta, S., Sané, Y., Vischel, T., and Camara, M.: Simple scaling of extreme precipitation regime in Senegal, Scientific African, 23, e02034, https://doi.org/10.1016/j.sciaf.2023.e02034, 2024.

Miller, J. E.: Basic concepts of kinematic-wave models, US Geological Survey, 1984.

Sy, B., Frischknecht, C., Dao, H., Consuegra, D., and Giuliani, G.: Reconstituting past flood events: the contribution of citizen science, Hydrol. Earth Syst. Sci., 24, 61–74, https://doi.org/10.5194/hess-24-61-2020, 2020.

Tadesse, A. and Anagnostou, E. N.: African convective system characteristics determined through tracking analysis, Atmospheric Research, 98, 468–477, https://doi.org/10.1016/j.atmosres.2010.08.012, 2010.

---

## Author Response (AR1)

**Response to the Reviewers Comments**

**Manuscript number** : egusphere-2023-2458 (NHESS)

**Article type** : Research Paper

**Article title:** Detection of flooding by overflows of the drainage network: Application to the urban area of Dakar (Senegal)

**Authors :** Laurent Pascal Malang Diémé, Christophe Bouvier, Ansoumana Bodian and Alpha Sidibé

Dear Reviewers,
Thank you for your insightful suggestions and comments on our manuscript. We have carefully reviewed your comments and incorporated the suggested changes. Throughout the document, the reviewers comments are written in blue, while the authors responses are shown in black. Please find below a summary of the key corrections made in the paper, in addition to our detailed responses to the reviewers' comments and remarks.

In this new version, we have reorganized the structure of the paper to make it easier to read and understand. We have extended section 2 to add a "Datasets" sub-section (L114) which provides more details on the data used (Geographic datasets, IDF curves), sources, and level of accuracy (in particular the DTM). Section 3 is devoted entirely to presenting the method (L142). Here, sub-sections 3.1 and 3.2 have been adjusted for clarification purposes. Section 4 (L302) remains unchanged and describes the calibration of the model. Section 5 (L424), which deals with the Modelling of the drainage overflow, has been reorganized into 3 successive sub-sections: Design storms construction (L426), Implementation to detect network overflows (L471), and Discussion (L494).

**Reviewer #1**

**General comments**

The presented study falls within the scope of NHESS. However, There are many points that should be clarified before considering the paper for publication.

We thank the reviewer for the valuable comments and appreciate the useful suggestions to improve the manuscript.

**1. Major Comments**

**Comment 1.** A flow chart of the methodology should be used to present the methodology. This will help the reader to understand the proposed modelling system

**Response :** We have inserted a flow chart of the methodology ( see Figure 2).

[Figure]

**Comment 2.** It is not clear to me whether the drainage system (i.e. stormwater drainage network and retention basins) is constructed or it is planned. It is strange to me that the drainage network is a network of open channels of orthogonal cross section. Stormwater drainage network is usually underground and consists of pipes. If the network exists then the dimensions are set and known otherwise the dimensions of the drainage network elements (i.e. pipes and cannals) is a matter of design. The authors should clarify this issue

**Response :** We have provided in section 2.2.1 a more detailed description of the stormwater channel structure, and retention basins (L121-125).

**Comment 3.** In continuation of the previous comment, more information about the study area should be presented, e.g. climate, historical extreme rainfall and flood events, hydrology, DEM, etc.

**Response :** We have extended the description of the study area to include relief, climate (L80-89), and the various flood situations that have affected the city (L101-105). In the same way, to provide a better description of the study area, we have also inserted a figure (Fig. 1) showing the distribution of elevations (Fig. 1b) and soil types (Fig. 1c). Information about Digital Terrain Model (DTM), are also presented in section 2.2.1 (L119, 120).

**Comment 4.** More information about the Kinematic Wave (KW) flow routing model should be given in Section 3.4. The governing equations of KW to be solved should be presented.

**Response :** We have inserted a description of the 1D kinematic wave model and its governing equations (L249-281)

**Comment 5.** Section 3.5. Why a simple linear storage model is not used for water retention structures?

**Response :** Adopting a linear reservoir behavior is indeed more suitable for simulating the emptying of retention basins. This principle has been explained more clearly in the new version of the paper. We have then rewritten sections 3.5 (L283-292 ) and 4.3 (L399-404)  for a better precision. We have also inserted table 2 which illustrates the principle of emptying.

**Comment 6.** All areas have the same soil characteristics as found in the experimental site. It would be more realistic to have a soil map of the area or CN maps to estimate the parameters of SCS rainfall abstractions (or effective rainfall) model.

**Response :** We made the necessary corrections, and inserted a soil map (Figure 1c) provided by the National Institute of Pedology of Senegal. We have also described the dominant soil types (L85-86). However, the soils appear very sandy all over the area, which does not require use different CN values according to the soil type. Landuse and urbanisation remain the only parameters explaining CN's spatial variability.

**Comment 7.** What is the basin response time (Tr)? Is it the time of concentration or the time lag? In Equation 11, please explain what is Tm (transfer time). Why Tr is not estimated by widely used common and typical equations?

**Response :** Tr is indeed a lag time, considered as the time between the center of gravity of the runoff and the center of gravity of the rainfall. The measurements of both rainfall and runoff in the experimental catchment of Fann-Mermoz led to an estimation Tr = 30 mn. As Tr can be also derived from the model under some simplified hypothesis, it allows to estimate Vo (the average velocity of the flow over the path traveled). This seems to be more reliable than empirical common equations. We have rewritten section Section 4.1.2 to give more detailed explanation of the method (L342-355).

**Comment 8.** Give the general equation of IDF curves as $i = CT^nD^{-k}$

**Response :** We inserted a new table (Table 1) showing rainfall of different durations (1, 2, 4, 6, 9, 12, 24h) and different return period (2, 10, 100 years).

**Comment 9.** How and why a 4-hour rainfall is selected? Is 4 hours the critical duration of a storm? Please explain.

**Response :** Indeed 4 hours is mostly the life duration of rainstorms generally observed in African convective systems as analysed by Tadesse and Anagnostou (2010). We have applied this total duration for Dakar (see L458-463), even though detailed local analyses are required to assess the structure of significant precipitation events. The former critical duration of rain that we extracted from the IDF provided by Sane et al. (2018) is 1 hour. For future work, possible improvements would be to choose the critical duration less than 1 hour. Integrating durations less than 1 hour has been tested using the IDF curves recently updated by Diedhiou et al. (2024) . New simulations with the model show that the project rainfall associated with a critical duration of 1 hour tends to underestimate peak flows by an average of 7% compared with the project rainfall associated with a critical duration of 30 mn, regardless of the surface area of the watersheds, ranging from 10 ha to 12 km². Furthermore, a 10 mn led to underestimate the peak flows associated to a 30 mn critical period by an average of 9%.

[Figure]

*Comparison of peak flow between an intense rainfall of 30 minutes and 1 hour with a total duration of 4 hours.*

**Comment 10.** Why the spatial distribution of design rainfall is not considered? The same design hyetograph is applied over the study area.

**Response :** The simulations were carried out on the assumption that the rainfall was uniform over all the defined catchment areas. Such hypothesis does not account for areal reduction factor, but can be adopted because most of the catchments (i.e. 94%) have small areas, less than 2 km$^2$. The largest catchment areas account for 4%(2 to 6km$^2$) and 2% (6 to 12km$^2$) out of the 890 catchment outlets which have been considered, as shown below. We have inserted it into the discussion (L514-524)

[Figure]

**Comment 11.** A major drawback of the study is that the methodology has not been validated against historical flood events. The results presented are purely theoretical and could be fictional and not representative. The authors should simulate one or two events for validating the method and the modelling system.

**Response :** Indeed, a key limitation of this study is the lack of validation of the simulation results, as it was pointed out in the discussion. Although it is of major importance, the validation task cannot be undertaken at the moment. However, we hope that the methodological aspects of the simulation should be of enough interest to be published. The validation task is a priority and a perspective for our future work (see L543-550). The sentence is as below:

"As things stand at present, it was not possible to get the necessary data for the validation of the method, which means on the one hand sub-daily rainfall data, and on the other hand flood maps for the recent events that occurred in Dakar.  The imminent installation of a rain gauge radar in Dakar could help to facilitate this.  Flood maps could be obtained by exploring citizen science tools (Sy et al., 2020) or ordering a high-precision satellite image to map out flooded areas."

**Comment 12.** Conclusions. The authors correctly write the deficiencies of the methodology but they should outlined and discussed these deficiencies earlier in the paper.

**Response :** We agree and have mentioned the limitations of the work (in particular the non-validation of the simulations) in the abstract section (L 26-27)

**Comment 13.** There are many awkward hydrological terms. Proper hydrological terms should be used. Some of them are indicated in the minor comments bellow.

**Response :** Thank for the careful review.

**Comment 14.** English language needs improvement. In some paragraphs, the English writing is poor.

**Response :** We agree and have carefully correct this aspect all over the manuscript

**2.  Minor comments**

**Comment 1.** *There many improper hydrological terms. For example:*
*a. Line 93. "…hydrological production …." Please revise to "…..flow generation…."*
**Response :** We removed "hydrological production" into "rainfall-runoff model " L145

*b. Line 97. "……injected in the model….." Please revise to "……used as input data to the model…"*
We changed "injected in the model" into "……used as input data to the model…" L149,  L427

*c. Line 199. "…production model…" Please revise to "….hydrological model…."*
we have accordingly revised "…production model…" into"….hydrological model…."  L310

*d. Line 308. "…..project…" Use the term "design"*
We changed "…..project…" into "design storm" L149, 426, 427,  443, 469, 479).

*And others.*
**Response :** We have reviewed the terms and corrected them accordingly all over the manuscript (L10, 14, 20, 31, 33, 34, 95, 147, …)

**Comment 2.** *Equation 5. Not "si". It is "if"*
 **Response :** Corrected (L243)

**Comment 3.** *Line 180 and elsewhere. What is the OC model? It has not been described.*

**Response :** All along the paper, OC has been changed to KW which is the used kinematic wave model (L274, 531, 533).

**Comment 4.** *Table 2. It is not understandable. Use the equation of reservoir level-storage volume outflow curves.*

**Response :** We have changed and rewritten the paragraph explaining the principle of the used linear reservoir model (L283-292  and L399-404) and given an example in the table by considering an emptying rate for the reservoirs that varies according to the water level in the reservoir (see Table 2).

**Reviewer #2**

**General comments**

This study aims to model a fine-scale run-off model of the urban area and assessment of the response of the storm drainage network (canals and retention basins) to different rainfall events. The methodological approach is based on a preliminary reconstruction of the drainage directions modified by urbanization and the implementation of combined hydrological and 1D hydraulic models calibrated to the city's urban conditions. My minor and major comments for this manuscript are as follows.

**Response :** Thank you for your valuable input. The manuscript has revised, taking into account all the issues you raised. Please find our detailed response below:

**1.  Minor comments**

Grammatical errors

**Comment 1.** Line 42 – 44: There is a repeated sentence here.

**Response :** We removed the repeated sentence (L48-49)

**Comment 2.** Figure 1. Please insert the north arrow into the map

**Response :** We have inserted the north arrow  into the map in Figure 1. We have also added an altitude map and a soil types map to provide a better description of the study area (see Fig 1b, c).

**Comment 3.** The title does not match the main objectives of this study. The authors should confirm the aim of this study focusing on the detection of floods or modelling of drainage networks.

**Response :** We agree and propose to change the title "Detection of flooding by overflows of the drainage network: Application to the urban area of Dakar (Senegal)" to "**Modelling urban stormwater drainage overflows for assessing flood hazards: Application to the urban area of Dakar (Senegal)**"

**Comment 4.** Please insert a flow chart of data processing

**Response :** We have inserted a flow chart of the method (see figure 2)

**Comment 5.** The study should integrate the validation of flood simulation results using accurate reference data.

**Response :** Indeed, a key limitation of this study is the lack of validation of the simulation results, as it was pointed out in the discussion. Although it is of major importance, the validation task cannot be undertaken at the moment. However, we hope that the methodological aspects of the simulation

should be of enough interest to be published. The validation task is a priority and a perspective for our future work (see L543-550)

**Comment 6.** Section 3.1. please give a brief explanation about ATHYS modelling that was applied in your study

**Response :** In addition to the internet link to the ATHYS platform, we have expended the description of ATHYS to provide more details (L 152 - 154). The sentence is as follows:

"The modelling chain was built in the ATHYS platform, developed by Hydrosciences Montpellier. ATHYS enables a range of hydrological and hydraulic GIS-based models (MERCEDES unit), as well as geographical (VICAIR unit) and hydrometeorological (VISHYR unit) data processors. ATHYS is a free software, available from www.athys-soft.org "

**Comment 7.** What is the source of the DTM data and soil map used in this study? Please explain in detail about the resolution as well as the accuracy of these data.

**Response :** A soil map and its source are provided in Figure 1c. As for the DTM we have provided new sub-section "Geographic datasets" for additional informations (See L119-120).

**Comment 8.** Line 155: how the model calculates the average velocity of the flow (Vo) is not specified.

**Response :** We have rewritten this paragraph to improve its clarity (see L342-355). The parameter Vo (velocity of the surface flow) was determined using data from the Fann-Mermoz gauged catchment.

**Comment 9.** Section 5 – Results and discussions part also does not match the main objectives of this study. Section 5.1 should robustly specify the results of flood simulation using the ATHYS modeling method as three methodological steps shown in the abstract.

**Response :** The discussion section was revised to better highlight the strengths, limitations and improvements of this work (L512-524). Additional details on the identified method validation strategies are also provided (L543-550).

**References**

Constantindes, C. A.: Numerical techniques for a two-dimensional kinematic overland flow model., Water SA, 7, 234–248, 1981.

Diedhiou, C. W., Panthou, G., Diatta, S., Sané, Y., Vischel, T., and Camara, M.: Simple scaling of extreme precipitation regime in Senegal, Scientific African, 23, e02034, https://doi.org/10.1016/j.sciaf.2023.e02034, 2024.

Sy, B., Frischknecht, C., Dao, H., Consuegra, D., and Giuliani, G.: Reconstituting past flood events: the contribution of citizen science, Hydrol. Earth Syst. Sci., 24, 61–74, https://doi.org/10.5194/hess-24-61-2020, 2020.

Tadesse, A. and Anagnostou, E. N.: African convective system characteristics determined through tracking analysis, Atmospheric Research, 98, 468–477, https://doi.org/10.1016/j.atmosres.2010.08.012, 2010.